# A Framework to Assess the Sustainability of Platform Economy: The Case of Barcelona Ecosystem

**Mayo Fuster Morell [1] and Ricard Espelt [2],*** 

[1]   Berkman Klein Center for Internet and Society, Harvard University, Cambridge, MA 02138, USA;
      mfuster@uoc.edu
[2]   Internet Interdisciplinary Institute, Open University of Catalonia, 08018 Barcelona, Spain
*   Correspondence: ricardespelt@uoc.edu

**Abstract:** This article presents a framework for evaluating the sustainability qualities of Platform Economy initiatives. It takes into account governance, economic model, technology, data policies, social responsibility and impact. The framework has been tested empirically in a sample of one hundred commons-based peer-to-peer production cases identified in Barcelona. Data collection was based on online ethnography and structured interviews. The results reveal the different levels and tendencies of pro-democratization. It appears that the cases that are more sustainable are also sustainable in other dimensions. The analysis found a correlation between governance and technology and data models, and it further demonstrated that governance is correlated with the economic model. Both results together indicate that the governance of a platform plays a central role in its overall approach.

**Keywords:** sustainability; democracy; governance; socio-economics; research

## 1. Introduction

The Platform Economy (PE) (also referred to as the sharing economy) is the propensity for trade to progressively move towards and favor digital platform business models. Businesses are facilitated by community-based online platforms that allow a wide range of human activities. The digital platforms open the way for radical change in how businesses work, trade, socialize and work together in the consumption and production of goods and capital. The PE is developing at an exponential rate, building substantial interest, and has become a top priority for governments across the world [1,2]. Yet, the PE experiences two fundamental difficulties: (1) The PE has positive environmental impacts and promotes a potential pathway to sustainable societies [3–5], constituting a paradigmatic change [6]. However, the PE lacks a holistic framework to assess its sustainability. For example, the sustainable design of the platform has examined economic and technical facets, but has not considered other significant sustainability measures including environmental effects, inclusion, gender, or legal ramifications, therefore, lacking an appropriate multidisciplinary perspective.(2) There is confusion about the PE which presents it as collaborative, in terms of open governance, open technologies and data, while actually other PE models also experience comparable ambiguities [7]. Unicorn extractionist corporation platforms including Airbnb and Uber are inciting great debate [1,7]. Other PE frameworks that are both collaborative and successful do exist, for example, platform cooperativism, open commons and decentralized organizational structures where decision-making authority is based on free knowledge and the social economy, yet researchers and policymakers have not acknowledged the importance of these alternative PE models. Furthermore, there is an absence of a framework that is able to compare, classify and contrast different PE models.

Overall, the PE creates paradigmatic change, but in order to redirect the PE towards a sustainable future, researchers need to focus on these two limitations.

The present work will produce a sustainability democratic quality balance of the PE to overcome the aforementioned challenges [8]. The quality balance enables researchers to analyze digital platforms and compare different PE models by exploring the PE initiatives in terms of their democratic and sustainable features. The quality balance examines the economic strategy, governance, technology, data policy, impact and social responsibility towards platform externalities.

## 2. A Quality Balance to Assess the Sustainability of the Platform Economy

The possibility of the PE to promote sustainable societies has been highlighted since its initial characterization [3–5]. Nonetheless, research surrounding the socio-economic and environmental impacts of the PE is ambiguous and inadequate. Less than 9% of the literature regarding the PE has investigated the potential benefits, costs and welfare impact of the PE [1].

Sustainability qualities of the PE have integrated economic, social and environmental sustainability dimensions [3]. The ex-ante analysis regarding its impact and effectiveness in terms of sustainability has examined environmental impacts, self-employed work, job stability and consumer welfare; however, researchers have not conducted a comprehensive analysis of the integration of sustainability into economy, community and social aspects [9].

Previous research investigating the sustainability and democratic qualities of PE initiatives remain limited and incomplete. In terms of the social aspect, Richardson (2015) regards PE sustainability as a determinant of change which contributes to the reduction of social inequalities [10–12]. Some studies argue that peer-to-peer activities, such as sharing the access to goods and services benefit those of below median income more than those of above median income. Some researchers go further and suggest that sharing companies can help redistribute income and reduce inequality. Schor's empirical work has detailed the importance of sharing-oriented platforms for promoting social connections and reducing ecological and economic concerns [13].

From an ecological viewpoint, Demailly et al. (2016) argue that in spite of the fact that sharing platforms and their users may be positively affected by sharing-oriented platforms, it can have negative effects, as instead of promoting collective consumption behavior, the PE can, in fact, encourage compulsive buying behavior, which is supported by empirical research [14,15]. Researchers suggest that better governance models are required to create shifts in collective consumption behavior However, sharing-oriented mobility, including the increase in ride-sharing, could influence travel behavior norms and contribute to restoring environmental and social demands. The multidisciplinary approach to sustainability is ideal, as it embraces the phenomenon's complexity; however, this approach is methodologically challenging [16].

Research strategies for investigating PE sustainability are often based on sustainability indicators adopted from corporate sustainability literature and readily available data [5]. But this research approach has several limitations. To start with, researchers do not agree on which sustainability indicators to utilize, and more often than not, the indicators are not suitable to the features of the PE (due to the activities being of a non-monetary nature), are focused on small scale entrepreneurs [17], and generate rebound effects which reduce the effectiveness of valuable contributions [18,19]. In regard to the PE's economic sustainability, current work has concentrated solely on the effect of unicorn models on vehicle-sharing [5,12,20], rental industries, tourism accommodation [21,22] and online labor [23,24].Additionally, research has focused on current incumbents and the negative impact of the unicorn model [25]. Besides, the stakeholders involved in such controversies are also involved in presenting the findings of the work. Thus, to determine the environmental effects of PE companies, researchers must thoroughly analyze their supply chains. Although companies like Airbnb and Uber have released an abundance of reports, information regarding the methodologies utilized and data obtained regarding their environmental effectiveness are not transparently outlined or

outwardly available to analysts and researchers [26,27]. Thus, the reliability of their claims cannot be independently validated.

In contrast, this research will reside with the sustainability in commons-oriented modalities study [28]. The study creates a framework of PE sustainability that aims to incorporate socio-economic, environmental, political, Internet and gender dimensions of sustainability. Compared to prior PE research, the three other democratic sustainability qualities: digital sustainability of the Internet as a commons, gender as a cause of inequality and political sustainability, will be considered. The Internet is a living ecosystem of common resources which need to be protected in regard to openness, decentralization and net neutrality, according to the net environmental approach [29,30], while the literature surrounding the PE refers to the Internet as an unchangeable source, just accommodating sharing-economy platforms. As part of the sustainability framework, the degree in which the models add to policy system quality and regulatory requirements has also been considered.

## 3. The Platform Economy's Multidisciplinary Balance

The sustainability qualities of the PE are articulated around five dimensions, as shown in Figure 1.

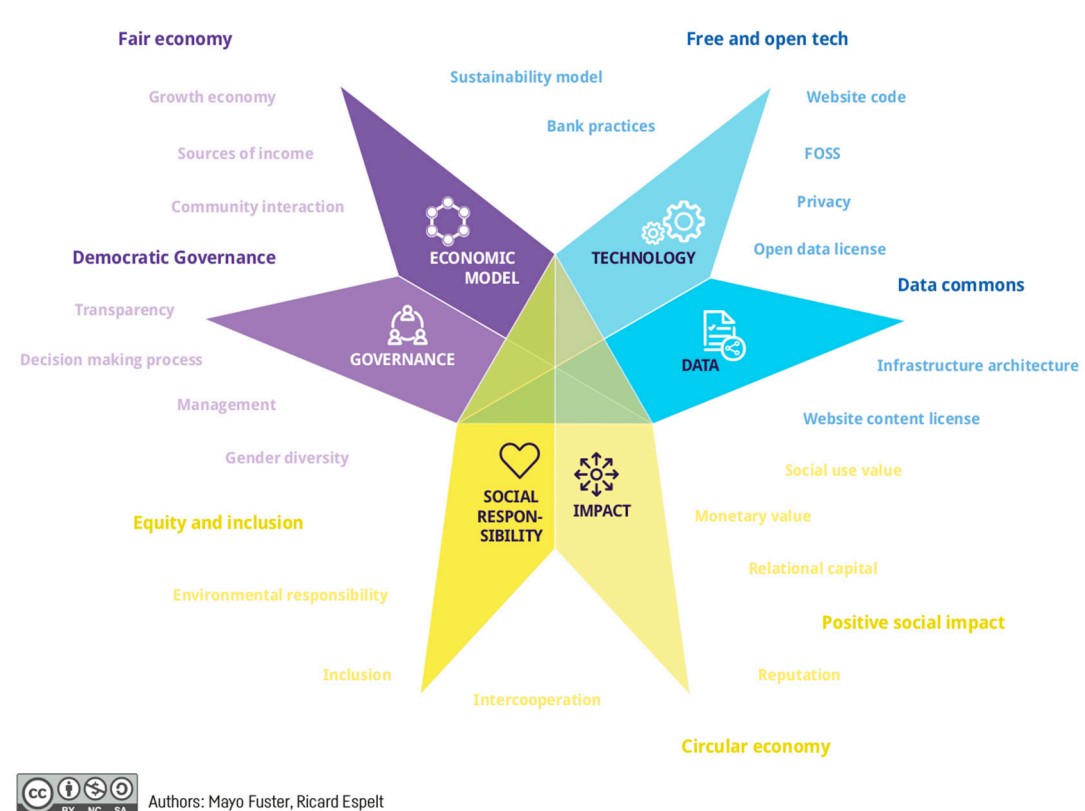

**Figure 1.** Star of sustainability qualities of digital platforms [7].

*3.1. Governance*

The degree to which an open modality was adopted by the platforms and different dimensions of platform governance were considered. The study evaluated the governance concerning platform provision (transparency, policies of participation and legal constitution) as well as governance at the platform interaction level (matching platform functionalities with the grade that users can participate) [31]:

(1)　In regards to the management of contributor openness, the study explored the following: (1.1) the manners in which the platform content is determined by users and whether creating new content or offering products and services only is possible; (1.2) platform participation policy: if platform participation is open without filters, moderated before publication, or moderated after publication; (1.3) user interaction: if users form groups or communicate among themselves; and (1.4) if the platform considers single type open access to any user or different types of user accounts [31].

(2)　In terms of election of administrator openness, the following were explored: if administrators are self-appointed or elected by the general community including fellow platform administrators; and if participation leads to the automatic gain of privileges for platform administrators.

(3)　Community interaction: whether community decision-making is aided by systems (formal or informal) and if platform policies and formal rules regarding community interaction are openly available.

(4)　The legal entity type and how the legal entity is interacting with by community members. Different legal entities including university, cooperative, association, business, company, foundation, public administration and without legal format, were considered.

(5)　In regard to economic management linked to governance, the following were explored: (5.1) economic transparency (is the economic balance provided publicly, or is it only available to the community); and (5.2) the openness of the decision on where the project benefits go to. For example, is it the community or just the owners of the project who are informed about and can manage such benefits.

## 3.2. Economic Model

The relationship between the economic benefits and their social impact, as well as the economic sustainability of the project, and their financial models were considered. The study evaluated:

(1)　Economic orientation, taking into consideration: (1.1) the legal entity type and the potential economic return that is established with the community in terms of its financial model (from more to less community): the different legal entities including university, cooperative, association, commercial company, foundation, public administration or without legal format; (1.2) economic benefit destination: whether the benefits are divided between proprietors or reinvested into the project; (1.3) whether the growth model is organic (economically escalates without influencing the governance model) or if the model is replicable, or speculative (is the model able to achieve maximum growth and does the project become a sellable asset); (1.4) whether monetary exchanges between users occur: almost always, often, sometimes, almost never or never; and (1.5) whether the platform uses banking services ethics or not.

(2)　In order to evaluate its sustainability, the study explored the initiative's economic balance.

(3)　The type of resources utilized in the platform's financial models will be studied, namely: public funding, external and internal non-monetary donations, organization of events, family savings, private capital, research programs (H2020), microfinance, commercialization of the brand and data, free resource, training programs, prizes, by-products, bank credit, quotas, advertising, premium services, alternative currencies and merchandising.

## 3.3. Technology Policies

The openness of technological policies refers to technological architecture and software that favors openness and freedom.

The type of license used by the platform has been used as an indicator. The ways in which the platform favors openness or "freedom" categorizes the type of platform license. In this study, robust licenses that allow freedom to be maintained from author to end-user, where copy that was left is used so other work is included under the same copyright as the original, such as Lesser General Public License (LGPL) and General Public License (GPL), were prioritized.

On the other hand, flexible licenses including permissive software licenses, like that of Berkeley Software Distribution (BSD) and the Massachusetts Institute of Technology (MIT), where the distribution of work is free or private, was placed. Alongside this, the study located platforms without a license or all rights reserved. In regard to technological architecture, the study adopted two indicators. First, whether the infrastructure of the technology is less open or more open, and also taking into consideration if the model is reproducible (source code as Free Libre Open Source Software—FLOSS availability) and its distribution from peer-to-peer to federated to centralized. The study considered: (1) Centralized but not reproducible, because one node is exclusively provided by the platform owner and proprietary (e.g., Facebook); (2) Centralized in one entrance point (e.g., Wikia); (3) Federated (e.g., Kune); (4) Centralized reproducible FLOSS, but not federated (e.g., Media wiki); and, (5) Peer-to-peer (e.g., BitTorrent) [31]. The use of blockchain (Yes/No) was also considered with the objective of decentralizing the technological architecture of the platforms and opening up community participation.

### 3.4. Data Policies

Two elements of platform policies were adopted: data and content. The content aspect refers to the type of user-generated content license. The license used and their categorization from less open to more were: (1) All rights reserved or No license; (2) CC BY-NC-ND; (3) CC BY-NC-SA; (4a) CC BY-ND; (4b) CC BY-NC; (5a) CC BY-SA; (5b) CC BY; (6) CC0; and, (7) Public domain. The possibility to keep the same license attributions (CC BY-SA) and to share only by author recognition (CC BY) were balanced.

In terms of data policy, the study adopted access to user-generated data as its indicator. The options explored were (from less to more open): (1) Not possible to export, copy, or access any application programming interface (API); (2) Freely downloadable in part; (3) API with some restrictions; (4) Freely downloadable as a whole; (5) Full data export (data dump); and, (6) API without restrictions [31].

### 3.5. Social Responsibility and Impact

Social responsibility and impact relate to awareness or responsibility toward the negative implications of the PE, including social inequalities and social exclusion. It also concerns equal access to the platform regardless of gender, social class and income. In addition to this, social responsibility and impact involves the common good of the city; the preservation of inhabitants' rights to the city; the impact of the PE in terms of the environment and policy; compliance with health and safety standards; and the protection of public space and human rights [32].

The sharing-oriented PE favors (1) peer relationships, in comparison to traditional hierarchical command and contractual relations, and the inclusion of a community of peers involved in platform governance; (2) value distribution and does not disguise profit motivation under the pretense of sustainability; (3) privacy-aware public infrastructure, that leads to open-access of commons resources that favor reproducibility; and, (4) the responsibility of platform externalities [27].

## 4. Methods

### 4.1. Participants and Design

The study utilized one hundred projects which included projects promoted by different types of actors (communities without legal format, public administration, cooperatives and companies,), in different areas (tourism, culture, mobility), goals (community engagement, business, knowledge co-creation) and non-profit and profit-oriented. Fifty platform managers contacted through the information retrieved on their website were interviewed to gain an insight into the characteristics of the platform and the organization behind it.

Due to the unsuitability of developing a probability sample of diverse digital platform experiences and the lack of adequate conditions and a comparability goal, non-proportional quota sampling was used to create the 100 case sample, which was narrowed down from an initial 1000 commons-based peer-to-peer production cases which were identified in Barcelona by the P2P value directory project [33].

Matching criteria was employed to ensure the diversity of the initial 1000 cases. Both platforms with global activity and local cases were considered during selection. This approach was utilized, as both global and local platforms develop their activity on the city spectrum.

Additionally, systematization of the sample was conducted to improve the sample's robustness. The 100 most relevant cases were selected in terms of: (1) Projects with a significant trajectory (not just started); (2) Projects based on the platform economy; (3) Projects supported by a digital platform; and lastly, (4) Projects based in Barcelona [33].

*4.2. Data Collection*

Data collection was based on a "codebook" (Codebook used for data collection http://dimmons.net/wp-content/uploads/2019/09/Full_Col_lacy_CODEBOOK.pdf).The following sustainability indicators were used: Governance, Economic Model, Technology Policies, Data Policies and Social Responsibility and Impact. Two methods of data collection were utilized: structured interviews and digital ethnography. To verify the data obtained from the main researcher, two other researchers tested the codebook indicators with random PE cases.

4.2.1. Digital Ethnography

The PE case data were collected through digital ethnography. The information was retrieved by surfing the Internet and the use of metric tools like Alexa (alexa.com) or Kred (home.kred).

4.2.2. Structured Interviews

Structured interviews with the managers of fifty of these one hundred PE cases were performed. The contact details of the managers were obtained through the platform website. The goal of the interviews was to delve deep into the model of the platform, specifically its economic model. Thus, the information gathered amplifies the web collection data. The structure of the interview followed the codebook indications (phone collection questions). The answers to the questions were collected via an online survey filled in by the same interviewer.

*4.3. Data Analysis*

Descriptive statistics and tests of normality of the defined variables in the codebook for each dimension were performed. Because the data were not normally distributed, bivariate non-parametric correlations among dimensions and the different subdimensions were conducted using Pearson's correlations. Pearson's correlations were employed to explore the relationships between open governance, knowledge openness variables and open technology.

## 5. Results

*5.1. Barcelona Platform Economy Ecosystem*

The analysis points to a diverse and plural Barcelona PE ecosystem regarding the geographical base of their headquarters and their community, year of creation, evolutionary stage, field of activity, digital dimension and type of interaction.

Barcelona was the principal headquarters of 64.36% of the cases analyzed, 16.83% of cases were located in other parts of Spain, 11.88% in Catalonia, 4.95% of cases were in Europe and 1.96% were in the rest of the world [33]. Focusing on the community, 42% of cases were international, while 20% were Spanish, 22% Catalan, 8% European and 8% were from Barcelona.

As shown in Figure 2, since 2010, 77.23% of the commons-based peer-to-peer production cases have been created. In terms of evolution, since 1995, 52.48% of platforms are considered mature, 22.77% are in a growing phase, 10.89% of cases have a stable and full operational mode, 9.90% of platforms have an early stage of business model implementation, and 3.96% of platforms are no longer in operation [33].

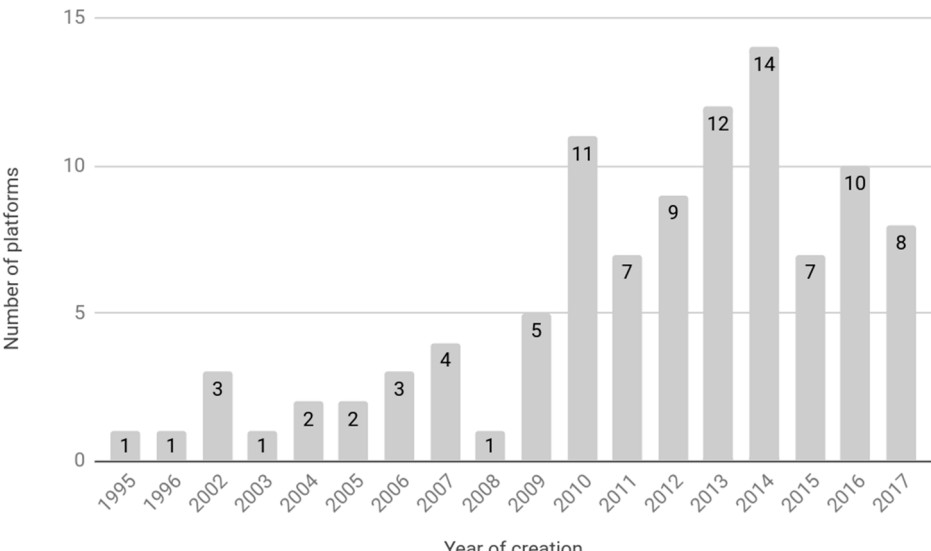

**Figure 2.** Year of creation.

In terms of the sharing-economy platforms' sector, 18.8% of cases are in the cultural sector, 13.9% are in the P2P economy and 10.9% in the mobility field (10.9%) [27], but there are various other sectors with PE qualities including tourism and housing and delivery services (Table 1).

**Table 1.** Percentage of sharing-oriented economy platforms regarding their area (*n* = 100).

| Area | % |
|---|---|
| Culture | 18.81% |
| P2P economy | 13.86% |
| Mobility | 10.89% |
| Recycling-Circular economy | 6.93% |
| Tourism and housing | 5.94% |
| Collaborative networks | 3.96% |
| Software | 3.96% |
| Delivery service | 3.96% |
| Food and/or agroecology | 3.96% |
| Open technology | 2.97% |
| Collaborative mapping | 2.97% |
| Telecommunications | 1.98% |
| Citizen participation | 1.98% |
| Leisure | 1.98% |
| Collaborative writing | 1.98% |
| Education and training | 1.98% |
| Design and makers | 1.98% |
| Co-working | 1.98% |
| Legal and labor assistance | 1.98% |
| Sensor networks | 0.99% |
| Textile and accessories | 0.99% |
| Health | 0.99% |
| Cleaning and care economy | 0.99% |
| Power | 0.99% |
| Gig economy | 0.99% |

In 74.3% of the projects, the focus of activity is based on digital interactions, compared to 25.7% whereby the sharing-oriented economy platform is a further support. As shown in Table 2, 44.6% of the cases focus on peer-to-peer interaction and 22.8% of the activity is between consumers.

**Table 2.** Percentage of sharing-oriented economy platforms regarding their area (*n* = 100).

| Type of Digital Interaction | % |
|---|---|
| P2P | 44.6% |
| C2C | 22.8% |
| B2C | 21.8% |
| B2B | 6.9% |
| P2B | 4.0% |

*5.2. Governance*

In terms of contributor management (Table 3), 42.6% of platforms focused on offering, demanding, or rating products or services, and in 39.6% of platforms, users created new ways of generating content with others. This is also evident as 57.4% of the platforms allowed participants to be part of groups and communicate freely among themselves. Thus, there appears to be a balance of platform and user contribution.

**Table 3.** Management of contributors.

| Management of Contributors | Type of Form | Percentage |
|---|---|---|
| G1. Openness to contribution on the digital platform (*n* = 100) | Creating new ways of adding content | 7.9% |
| | Creating contents with others | 31.7% |
| | Offering, demanding and rating products or services | 42.6% |
| | N/A | 17.8% |
| G2. Policy of platform participation (*n* = 100) | Publication without filters | 35.6% |
| | Moderated previous publishing | 25.7% |
| | Moderated after publishing | 2.0% |
| | N/A | 36.6% |
| G3. Users can be part of groups and/or communicate among them (*n* = 100) | Yes | 57.4% |
| | No | 24.8% |
| | N/A | 17.8% |
| G4. Different types of account with diverse levels of permission (*n* = 50) | No | 28% |
| | Yes | 60% |
| | N/A | 12% |
| G5. Administrators' election (*n* = 50) | Self-appointed | 28% |
| | Privileges gained automatically by participation | 2% |
| | Elections among the general community | 2% |
| | By other administrators | 4% |
| | Selected by infrastructure provider with mechanisms of community representation | 2% |
| | Selected by the infrastructure provider | 30% |
| | Historical role (star) | 2% |
| | Selected by founders/leaders/board | 12% |
| | N/A | 18% |

In addition to this, most platforms did not restrict user participation, as members often published without filters (35.6%), and content moderation was only employed in 25.7% of the platforms. Despite this, 60% of platforms had different types of accounts with diverse levels of permission, and the administrators were selected by the platform providers or founders rather than the general community or self-appointment.

Regarding the decision-making process for community interactions (Table 4), in 54% of the projects, users are able to actively participate in defining the formal policies and rules. Whereas, in the remaining 56% of platforms, the community are involved in the decision-making through formally or informally defined systems.

**Table 4.** Community decision-making (*n* = 50).

| Community Decision-Making | Type of Form | % |
|---|---|---|
| G6. Decision-making systems in place for the community | Yes, formally defined | 50.0% |
| | Yes, informally defined | 6.0% |
| | No | 40.0% |
| | N/A | 4.0% |
| G7. Users can participate in the definition of formal rules and policies | Yes | 54.0% |
| | No | 34.0% |
| | N/A | 12.0% |

Regarding the legal entity (G8), the majority of platforms legally belonged to a business structure such as SCP, SL and SA for example (44.6%), while in 17.8% of the platforms, non-profit associations held legal rights and obligations, cooperatives were the legal entity in 12.9% of the platforms, public administrators had legal standing in 4% of the platforms and in 2% of the platforms the university was the legal entity. However, 13.9% of platforms did not have a defined legal format.

As demonstrated in Table 5, 76% of platforms make their economic balance accessible to members of the legal entity and 38% of platforms share this information publicly. In terms of the decision of the destination of economic benefits, 50% of platforms laid this decision with platform owners only, and 40% involved all the platform members.

**Table 5.** Governance linked to economic management (*n* = 50).

| Governance Linked to Economic Management | Type of Form | % |
|---|---|---|
| G9. Decision of the platform's economic benefits | The whole members | 40.0% |
| | Platform owners | 50.0% |
| | N/A | 10.0% |
| G10. Economic balance accessible to the members of the legal entity | Yes | 76.0% |
| | No | 16.0% |
| | N/A | 8.0% |
| G11. Economic balance being provided publicly | Yes | 38.0% |
| | No | 46.0% |
| | N/A | 16.08% |

As shown in Table 6, governance correlations highlight connections between the variables studied. First, the study found a strong positive correlation between legal entity type (G8) and user openness to contribute to the digital platform (G1)(r = 0.74, *p* < 0.01).A relationship was also observed between how the platform administrators are elected (G5),whether users can participate in the definition of the platform's rules and policies (G7),(r = 0.76, *p* < 0.01) and who makes decisions regarding the destination of economic benefits (G9) (r = 0.66, *p* < 0.01). Finally, a moderate correlation was found between the platform's legal entity and who controls the destination of economic benefits (G9) and economic balance transparency (G11) (r = 0.0.56, *p* < 0.05).

**Table 6.** Governance openness correlations ($n$ = 50).

|      | G1       | G2     | G3      | G4     | G5     | G6     | G7      | G8      | G9     | G10    | G11  |
|------|----------|--------|---------|--------|--------|--------|---------|---------|--------|--------|------|
| G1   | 1.00     |        |         |        |        |        |         |         |        |        |      |
| G2   | −1.00    | 1.00   |         |        |        |        |         |         |        |        |      |
| G3   | 0.45 *   | 0.39   | 1.00    |        |        |        |         |         |        |        |      |
| G4   | −1.00    | 0.28   | −0.11   | 1.00   |        |        |         |         |        |        |      |
| G5   | 1.00     | −1.00  | 0.34    | 0.23   | 1.00   |        |         |         |        |        |      |
| G6   | 0.31     | −1.00  | 0.53    | −0.04  | 0.41   | 1.00   |         |         |        |        |      |
| G7   | 0.33     | −1.00  | −0.11   | −0.04  | 0.76 **| 0.60 * | 1.00    |         |        |        |      |
| G8   | 0.74 **  | 1.00   | 0.27    | −0.03  | 0.70   | 0.58   | 0.55    | 1.00    |        |        |      |
| G9   | 0.16     | −0.07  | 0.35    | −0.18  | 0.63 * | 0.59 * | 0.66 ** | 0.73 ** | 1.00   |        |      |
| G10  | 0.27     | −1.00  | −1.00   | 0.11   | 1.00   | 0.46   | 0.71 *  | 0.61 *  | 0.61 * | 1.00   |      |
| G11  | 1.00     | −0.02  | 0.11    | −0.18  | 0.60 * | 0.29   | 0.55 *  | 1.00 ** | 0.56 * | 0.60 * | 1.00 |

**. Correlation is significant at the 0.01 level (2-tailed). *. Correlation is significant at the 0.05 level (2-tailed).

*5.3. Economic Model*

The PE has a rich and varied universe. Regarding the legal format, as displayed in Table 7, a balance between organizations that have a more community character (public administration 4%, university 2%, association 17.8%, foundation 5%, cooperative 12.9%) and one more character business (44.6%) was observed. But, 13.9% of the platforms did not have a defined legal format.

**Table 7.** Type of legal entity ($n$ = 100).

| Legal Format          | %     |
|-----------------------|-------|
| Public administration | 4.0%  |
| University            | 2.0%  |
| Association           | 17.8% |
| Foundation            | 5%    |
| Cooperative           | 12.9% |
| Commercial Company    | 44.6% |
| Not defined           | 13.9% |

The vast majority of projects (80%) reinvest the benefits that are generated, while only 2% of the initiatives divide the profits among the owners, or have other approaches to the economic benefits (Table 8).

**Table 8.** Destination of benefits ($n$ = 50).

| Destination of Benefits | %    |
|-------------------------|------|
| Reinvest                | 80%  |
| Divide among the owners | 2.0% |
| Others                  | 10%  |
| No answer/do not know   | 5%   |

In regard to the growth model, the majority of the projects is through an organic model (58%), while 20% have a reproductive model and 4% speculative (Table 9).

**Table 9.** Growth model (*n* = 50).

| Growth Model | Percentage of Use |
|---|---|
| Organic | 58% |
| Reproductive | 20% |
| Speculative | 4% |
| Others | 14% |
| No answer/do not know | 4% |

More than half of the projects studied (52%) do not encourage economic exchange among its members, while 8% do it almost never, 6% sometimes, 14% often and 16% almost always (Table 10).

**Table 10.** Economic interactions (*n* = 50).

| Economic Interaction | Percentage of Use |
|---|---|
| Never | 52% |
| Almost never | 8% |
| Sometimes | 6% |
| Often | 14% |
| Almost often | 16% |
| No answer/do not know | 4% |

Regarding the use of ethical banking services (Table 11), 40% of the initiatives are involved, 26% are not and 34% do not know, or if they have a legal format, they cannot use banking services.

**Table 11.** Ethical banking services (*n* = 50).

| Ethical Banking Services | Percentage of Use |
|---|---|
| Yes | 40% |
| Not | 26% |
| No answer/do not know | 34% |

Regarding the sustainability of the project (Table 12), 44% of projects indicate a positive economic balance, while 24% is negative and 32% do not know or do not value it.

**Table 12.** Economic balance (*n* = 50).

| Economic Balance | Percentage of Use |
|---|---|
| Positive | 44% |
| Negative | 24% |
| No answer/do not know | 32% |

Regarding the model of project financing (Table 13), we note that the most commonly used group of models (internal non-monetary donations 70%, public funding 64%, non-monetary external donations 58%, free resources 54%, organization of events 44%, microfinance 44%, training programs 42%, research programs (H2020) 38%, membership fees 30% and alternative currencies 28%) has a level of use (in 236 occasions) superior to the group of models of lesser democratic character (60% by-products or derivatives, 48% private capital, 42% prizes, 42% savings of relatives, 40%premium services, 32% brand marketing, 26% merchandising, 26% bank loans, 22% donations, 22% advertising and 12% commercialization of data).

**Table 13.** Financing model ($n = 50$).

| Financing Model | Percentage of Use |
| --- | --- |
| Non-monetary internal donations | 70% |
| Public funding | 64% |
| Non-monetary external donations | 58% |
| Free sources | 54% |
| Organization of events | 44% |
| Microfinance | 44% |
| Training programs | 42% |
| Research programs—H2020 | 38% |
| Membership fees | 30% |
| Alternative currencies | 28% |
| By-products or derivatives | 60% |
| Private capital | 48% |
| Prizes | 42% |
| Savings of relatives | 42% |
| Premium services | 40% |
| Brand marketing | 32% |
| Bank loans | 26% |
| Merchandising | 26% |
| Advertising | 22% |
| Monetary donations | 22% |
| Data commercialization | 12% |

Economic dimension correlations (Table 14) show that there is a moderate correlation between having a positive balance (E6) and the type of legal entity (E1) ($r = 0.57$, $p < 0.05$). Likewise, the type of entity (E1) favors that one of the financing models is participation in research projects H2020 (E11) ($r = 0.27$, $p < 0.01$), microfinance platforms (E13) ($r = 0.76$, $p < 0.01$), alternative currencies (E20) ($r = 1$, $p < 0.01$), non-monetary internal donations (E21) ($r = 0.71$, $p < 0.01$) or external (E22) ($r = 0.57$, $p < 0.05$). It is also observed that there is a strong negative correlation between the reinvestment of money in the project itself (E2) and the use of private equity for financing (E7) ($r = –0.65$, $p < 0.05$), which, on the other hand, is strongly related to obtaining a balance sheet positive economic (E6) ($r = 0.70$, $p < 0.01$). In this sense, there is also a strong correlation between using brand marketing (E15) as a model of income, and obtaining a favorable economic balance (E6) ($r = 0.61$, $p < 0.05$). In any case, the most important aspect of the economic dimension is the large number of correlations between the financing models.

**Table 14.** Correlations of the economic dimension of the PE initiatives.

| | E1 | E2 | E3 | E4 | E5 | E6 | E7 | E8 | E9 | E10 | E11 | E12 | E13 | E14 | E15 | E16 | E17 | E18 | E19 | E20 | E21 | E22 | E23 | E24 | E25 | E26 | E27 |
|---|---|---|---|---|---|---|---|---|---|---|---|---|---|---|---|---|---|---|---|---|---|---|---|---|---|---|---|
| **E1** | 1.00 | | | | | | | | | | | | | | | | | | | | | | | | | | |
| **E2** | 0.22 | 1.00 | | | | | | | | | | | | | | | | | | | | | | | | | |
| **E3** | 1.00 | −1.00 | 1.00 | | | | | | | | | | | | | | | | | | | | | | | | |
| **E4** | 0.40 | −0.39 | 1.00 | 1.00 | | | | | | | | | | | | | | | | | | | | | | | |
| **E5** | 0.33 | 0.23 | 0.19 | 0.02 | 1.00 | | | | | | | | | | | | | | | | | | | | | | |
| **E6** | 0.57* | 0.12 | 1.00 | 0.16 | 0.43 | 1.00 | | | | | | | | | | | | | | | | | | | | | |
| **E7** | −0.48 | −0.65* | −1.00 | −0.21 | −0.35 | 0.70** | 1.00 | | | | | | | | | | | | | | | | | | | | |
| **E8** | 0.46 | 0.15 | −0.30 | −0.55 | 0.21 | −0.29 | 0.39 | 1.00 | | | | | | | | | | | | | | | | | | | |
| **E9** | 0.13 | 0.39 | −1.00 | −0.40 | 0.26 | −0.29 | 0.08 | 0.98** | 1.00 | | | | | | | | | | | | | | | | | | |
| **E10** | 0.30 | −0.36 | −1.00 | −0.06 | 0.33 | 0.11 | 0.35 | 1.00** | 0.44 | 1.00 | | | | | | | | | | | | | | | | | |
| **E11** | 0.27** | 0.42 | 1.00 | −0.07 | 0.18 | 0.19 | 0.15 | 0.77** | 0.46* | 0.73** | 1.00 | | | | | | | | | | | | | | | | |
| **E12** | 0.27 | 0.02 | −1.00 | 0.32 | 0.03 | −0.16 | 0.27** | 0.82** | 0.56* | 0.80** | 0.62** | 1.00 | | | | | | | | | | | | | | | |
| **E13** | 0.76** | 0.23 | 1.00 | −0.05 | 0.14 | 0.51 | −0.01 | 0.69** | 0.65** | 0.60** | 0.77** | 0.60** | 1.00 | | | | | | | | | | | | | | |
| **E14** | 0.17 | −0.27 | −0.34 | −0.05 | 0.21 | 0.17 | 0.55* | 0.56** | 0.43 | 0.37 | 0.23 | 0.31 | 0.12 | 1.00 | | | | | | | | | | | | | |
| **E15** | 0.05 | 0.19 | −0.26 | −0.34 | 0.07 | 0.61* | 0.58** | 0.78** | 0.62** | 0.68* | 0.58** | 0.65** | 0.33 | −0.51* | 1.00 | | | | | | | | | | | | |
| **E16** | 0.22 | −0.44 | 1,00 | −0.09 | −0.11 | −0.51 | 0.65** | 0.82** | 0.68** | 0.46 | 0.52 | 0.59* | 0.56* | −0.69** | 0.77** | 1.00 | | | | | | | | | | | |
| **E17** | 0.04 | 0.13 | −0.04 | −0.02 | −0.29 | −0.40 | 0.68** | 0.62** | 0.41 | 0.26 | 0.23 | 0.46* | 0.18 | 0.20 | 0.67** | 0.62** | 1.00 | | | | | | | | | | |
| **E18** | −0.31 | −0.27 | 1.00 | −005 | −0.32 | −0.26 | 0.69** | 0.56** | 0.43 | 0.58 | 039 | 0.47* | 0.45 | 0.65** | 0.77** | 0.69** | 0.51* | 1.00 | | | | | | | | | |
| **E19** | 0.05 | 0.26 | 0.20 | −0.36 | 0.07 | −0.32 | 057** | 0.70** | 0.49 | 0.27 | 0.45 | 0.58* | 0.45 | 0.19 | 0.84** | 0.58** | 0.40 | 0.82** | 1.00 | | | | | | | | |
| **E20** | 1.00** | 1.00 | 1.00 | −0.27 | 0.43 | 0.56 | −0.04 | 0.74** | 0.67** | 0.38 | 0.63** | 0.59* | 0.65** | 0.61** | 0.69** | 0.62** | 0.20 | 0.47 | 0.71** | 1.00 | | | | | | | |
| **E21** | 0.71** | 0.05 | 0.30 | 0.27 | 0.14 | 0.44 | 0.04 | 0.36 | 0.45 | 0.24 | 0.63** | 0.57* | 0.83** | 0.50 | 0.21 | 0.57 | −0.06 | 0.50 | 0.47 | 1.00** | 1.00 | | | | | | |
| **E22** | 0.57* | 0.17 | 0.20 | 0.40 | 0.18 | 0.40 | −0.08 | 0.52* | 0.52 | 0.24 | 0.56 | 0.48* | 0.85** | 0.63 | 0.25 | 0.52 | −0.15 | 0.44 | 0.49 | 0.45 | 0.93** | 1.00 | | | | | |
| **E23** | 0.00 | 0.10 | −1.00 | −0.21 | 0.26 | −0.33 | 0.37 | 0.72** | 0.32 | 0.67** | 0.60** | 0.75** | 0.39 | 0.47 | 0.44 | 0.23 | −0.16 | 0.66* | 0.53* | 0.34 | 0.38 | 0.45* | 1.00 | | | | |
| **E24** | −0.28 | 0.16 | −0.07 | −0.42 | −0.06 | 0.00 | 0.68** | 0.62** | 0.52* | 0.52* | 0.58** | 0.68** | 0.42 | 0.36 | 0.88** | 0.62** | 0.16 | 1.00** | 0.75** | 0.60* | 0.52* | 0.36 | 0.53* | 1.00 | | | |
| **E25** | 0.00 | 0.13 | −1.00 | −0.34 | 0.18 | 0.00 | −0.40 | 0.77** | 0.57** | 0.73** | 0.72** | 0.75** | 0.59** | 0.07 | 0.70** | 0.24 | 0.47* | 0.55* | 0.57* | 0.38 | 0.49 | 0.44 | 0.71** | 0.84** | 1.00 | | |
| **E26** | −0.08 | −0.23 | −1.00 | 0.24 | −0.03 | −0.40 | 0.58** | 0.61** | 0.53* | 0.70** | 0.34 | 0.60** | 0.50* | 0.52 | 0.51* | 0.62** | 0.61** | 1.00** | 0.60* | 0.42 | 0.30 | 0.47* | 0.72** | 0.61** | 0.57 | 1.00 | |
| **E27** | 1.00 | −0.26 | 1.00 | −0.04 | 0.01 | −1.00 | 1.00** | 0.60** | 0.40 | 0.34 | −1.00** | 1.00** | 0.54 | −0.67** | 0.57* | 0.79** | 0.34 | 0.67** | 0.72** | 0.54* | 1.00 | 0.39 | 0.42 | 0.34 | 0.37 | 0.20 | 1.00 |

**. Correlation is significant at the 0.01 level (2-tailed). *. Correlation is significant at the 0.05 level (2-tailed).

## 5.4. Technology

Focusing on software openness (Table 15), 36.63% of the platforms adopted one of the different types of free licenses, 33.66% of the platforms use copyrighted software, whereas19.80% of the platforms use software without a license and 2.97% of these platforms use a public domain license to take advantage of software.

**Table 15.** Software openness (*n* = 100).

| Type of License | Percentage of Use |
|---|---|
| Public domain | 2.97% |
| CC BY-SA 3.0 | 3.96% |
| GPLv2 | 10.89% |
| GPLv3 | 3.96% |
| AGPL | 3.96% |
| LGPL | 4.95% |
| MIT license | 4.95% |
| Open Source License | 3.96% |
| All rights reserved | 33.66% |
| No license | 19.80% |
| N/A | 6.93% |

In terms of the openness of technological architecture and infrastructure (Table 16), the majority of the platform models were not open to being reproduced (44.55%). But 35.64% were open to reproducibility. Of the 35.64%, 18.8% of the platforms used centralized reproducible FLOSS, 10.89% had peer-to-peer architecture and 5.94% of the digital platforms used centralized FLOSS.

**Table 16.** Technological architecture openness (*n* = 100).

| Type of Architecture | Percentage of Use |
|---|---|
| Peer-to-peer | 10.89% |
| Centralized reproducible (FLOSS) | 18.81% |
| Centralized FLOSS | 5.94% |
| Not reproducible | 44.55% |
| N/A | 19.80% |

In addition to this, 44% of the platforms were not interested in using a blockchain to decentralize the platform's technological architecture and open up community participation. Yet, 38% of the platforms are already using or plan to use a blockchain. Overall, 39.6% of the platforms used free license software, 38% were interested in exploring other decentralized technology options and 35.64% of the projects were based on an open architecture. Technological open practices were utilized in only a third of cases.

The Pearson correlations (Table 17) demonstrated an association between the type of infrastructure architecture and platform software. A correlation between the use of blockchain technology and the openness of the platform code was also observed, although bearing lower statistical significance. Thus, it appears that different types of technological openness practices reinforce each other.

**Table 17.** Technological openness correlations (*n* = 50).

| Technological Openness | Open Software | Open Architecture | Block Chain |
|---|---|---|---|
| **Open Software** | 1.00 | | |
| **Open Architecture** | 0.93 ** | 1.00 | |
| **Block Chain** | 0.52 * | 0.56 | 1.00 |

**. Correlation is significant at the 0.01 level (2-tailed). *. Correlation is significant at the 0.05 level (2-tailed).

*5.5. Data Policies*

In terms of user-generated content (Table 18), the most commonly held license was the private copyright license (36.63%), followed by Creative Commons licenses (32.67%) and a small minority of platforms held a public domain license (2.97%). However, 23.76% of platforms do not hold any license.

**Table 18.** Knowledge content openness (*n* = 100).

| Type of License | Percentage of Use |
|---|---|
| Public domain | 2.97% |
| CC BY | 7.92% |
| CC BY-SA | 11.88% |
| CC BY-NC | 7.92% |
| CC BY-ND | 1.98% |
| CC BY-NC-SA | 2.97% |
| Copyright | 36.63%° |
| No license | 23.76% |
| N/A | 3.96% |

As shown in Table 19, 20.79% of platforms allow the exportation of user data.

**Table 19.** Data export openness (*n* = 100).

| Type of Data Exportation | Percentage of Use |
|---|---|
| Application programming interface (API) without restrictions | 5.94% |
| Free downloadable in whole | 10.89% |
| API with some restrictions | 1.98% |
| Free downloadable in part | 1.98% |
| Not possible to export, copy or API access | 53.47% |
| N/A | 25.74% |

Correlational analyses involving knowledge policies highlights an association between how the data is exported and user-generated content licenses, as shown in Table 20.

**Table 20.** Knowledge openness correlations (*n* = 50).

| Knowledge Openness | Content License | Data Export |
|---|---|---|
| **Content License** | 1.00 | |
| **Data Export** | 0.74 ** | 1.00 |

** Correlation is significant at the 0.01 level (2-tailed).

*5.6. Social Responsibility and Impact*

The majority of SE platforms (36%) indicated that there are a greater number of men than women using the platform. As shown in Table 21, in terms of the main indicators that characterize the social responsibility and the impact of the projects, 70% of the digital platforms favor the consumption of products or local services, 66% favor cooperation with other initiatives, 50% practice the circular economy, 40% of the platforms favor the inclusion of a community of peers despite the social exclusion risk and 20% have an initiative that favors positive environmental impacts.

**Table 21.** Social responsibility evaluation indicators.

| Social Responsibility Impact | Percentage |
|---|---|
| Social inclusion | 40% |
| Cooperation | 66% |
| Environmental responsibility | 20% |
| Circular economy | 50% |
| Local consumption | 70% |

Regarding the social responsibility and impact correlations (Table 22), the results demonstrate a moderate correlation (r = 0.67, *p* < 0.01) between the circular economy (SR4) and local consumption (SR5). At the same time, a full correlation (r =1, *p* < 0.01) between community participation (IMP2) and registered accounts (IMP3), and a strong correlation (r = 0.78, *p* < 0.01) between registered accounts (IMP3) and actively (IMP4). Finally, there was a moderate correlation (r = 0.51, *p* < 0.05) between the circular economy (SR4) and mission accomplishment (IMP1).

**Table 22.** Social responsibility and impact correlations (*n* = 50).

| | SR1 | SR2 | SR3 | SR4 | SR5 | IMP1 | IMP2 | IMP3 | IMP4 |
|---|---|---|---|---|---|---|---|---|---|
| **SR1** | 1.00 | | | | | | | | |
| **SR2** | −0.16 | 1.00 | | | | | | | |
| **SR3** | −0.47 | 0.08 | 1.00 | | | | | | |
| **SR4** | −0.09 | 0.11 | 0.33 | 1.00 | | | | | |
| **SR5** | 0.23 | 0.21 | 0.37 | 0.67 ** | 1.00 | | | | |
| **IMP1** | 0.01 | 0.25 | 0.07 | 0.51 * | −0.32 | 1.00 | | | |
| **IMP2** | 0.21 | 0.21 | −0.30 | 0.00 | −0.02 | −0.13 | 1.00 | | |
| **IMP3** | −0.04 | 0.04 | −0.39 | −0.04 | −0.12 | −0.15 | 1.00 ** | 1.00 | |
| **IMP4** | 0.18 | 0.31 | −0.54 | 0.26 | −0.01 | −0.16 | 1.00 ** | 0.78 ** | 1.00 |

SR1: Gender, SR2: Inclusion, SR3: Cooperation, SR4: Circular economy, SR5: Local Consumption, IMP1: Mission accomplishment, IMP2: Community participation, IMP3: Registered accounts, IMP4: Active contributions. **. Correlation is significant at the 0.01 level (2-tailed). *. Correlation is significant at the 0.05 level (2-tailed).

*5.7. Dimensions Correlations*

The strongest correlations among all dimensions (Table 23) from the analysis are between data and technology and between governance and economic model (r = 0.46, *p* < 0.01). On the one hand, this implies that there are relationships between the adoption of open data policies and open technology by platforms. The correlations between open governance and technological openness (r = 0.36, *p* < 0.01) and governance openness and data openness (r = 0.38, *p* < 0.01), demonstrates the importance of platforms adopting an open governance model and technological policy openness. On the other hand, it reinforces the connection between the governance of the platform and its economic model. At the same time, we observe a positive trend of data openness to reinforce a pro-common economic model (r = 0.31, *p* < 0.05).

**Table 23.** All dimensions correlations (*n* = 50).

| | Governance | Economic Model | Technology | Data | Social Responsibility and Impact |
|---|---|---|---|---|---|
| **Governance** | 1.00 | | | | |
| **Economic model** | 0.46 ** | 1.00 | | | |
| **Technology** | 0.36 ** | 0.20 | 1.00 | | |
| **Data** | 0.38 ** | 0.31 * | 0.46 ** | 1.00 | |
| **Social responsibility and impact** | −0.09 | −0.05 | 0.11 | −0.11 | 1.00 |

**. Correlation is significant at the 0.01 level (2-tailed). *. Correlation is significant at the 0.05 level (2-tailed).

## 6. Discussion and Conclusions

The present research has provided a framework of democratic sustainability qualities of PE to assess its sustainability. It takes into account governance, economic model, technology, data policies, social responsibility and impact. The framework has been tested empirically in 100 commons-based peer-to-peer production cases identified in Barcelona.

The framework of democratic sustainability qualities aims to address the current challenges of the PE regarding the lack of analytical tools to distinguish models and analyses sustainability and impact. In order to address these challenges, in the present work, the quality balance tool was employed to categorize the platforms and compare and contrast different PE models by examining their democratic sustainability qualities to provide greater understanding of sharing-oriented platform sustainability from several different perspectives. The results suggest the relevance of the dimension considered. It was observed in approximately one-third of the sample that democratic sustainability qualities in platforms are neither irrelevant nor prevalent. The cases which were more democratic in one dimension were less democratic in other dimensions. This finding implies that one aspect of a platform's ecosystem can be characterized as more democratic, while a larger segment is not based on any openness methods. The results revealed an association between the democratic sustainability indicators that define technological and knowledge policies, which are also linked with the economic model and governance. For example, the study discovered that democratic openness in data and technology are also reflected in other economic and governance models.

The analysis result of each of the dimensions also provides interesting insights. Regarding governance, it was observed that the majority of platforms allow users' participation, publishing without many constraints and facilitating the creation of groups in order to promote new content or rate, and demand products and services. In addition, most of the platforms, with different legal entities, involve the community in the decision-making processes. Furthermore, the majority of platforms allow their members to access the economic balance, and some platforms have spaces where the community can decide the destination of benefits. The correlation between the subdimensions of governance demonstrates how important the type of entity is in the way that contributors are managed, and their role in the destination of economic benefits.

Focusing on the economic model, the PE has a rich and varied universe, balancing organizations that have a more pro-public community character and more private and pro-market ones. In spite of that, the majority of platforms do not encourage economic exchange, reinvest their benefits and do not have a speculative approach. At the same time, ethical banking services, public funding and non-monetary donations have a great role in the model of sustainability. Thus, the voluntary work associated with the central role of the community is the main capital for the sustainability of the projects [27].

On the other hand, public policies are fundamental, since approximately two out of three projects receive funding from the public. Barcelona City Council, for example, has supported some projects through match-funding campaigns, whereby projects obtain sources from the community and public administration. While traditional models of funding (bank loans, merchandising, advertising, donations, etc.) have less presence, some new types of businesses, like the commercialization of data, have hardly been explored by researchers. In relation to internal economic correlations, we observe how legal entity impact in the model of funding and the large interactions among the different types of funding.

Focusing on technology policies, the majority of platforms are private, but open licenses are also represented. There are two possible explanations for this result. First, the restriction of website software use to platform owners only. The second explanation for platform privacy is the lack of attention to content licenses, software and open data exportation. In the same sense, technological architecture balances open and closed models, while projects are exploring blockchain, especially those which promote open code. At the same time, data policies replicate private licenses contents and non-downloadable data domination.

In regard to social responsibility and impact, even though most of the platforms have social and ecosystem responsibility, considering inclusion or collaborations with other actors (focusing on local) of their sector, the gender gap is sizable, and environment attendance is dismissed. Correlations show a great connection between the size of the community and their active participation.

One of the main observations is the key role of the platform governance model, which has a strong correlation with the economic model and technological and data policies (which are also correlated between them). Thus, it is concluded that the more democratic the platform governance is, the more democratic the platform's economic model will be. The research and analysis conducted to examine this connection have provided further support by reinforcing this relationship, particularly in terms of community involvement and participation. Therefore, in order to generate sharing economy platforms which improve both economic and environmental efficiency, democratic platform governance and transparency is the key. Another major conclusion regarding dimensions interactions is the disconnection between social responsibility and impact dimension with the rest of the dimensions analyzed. The traditional disconnection between open commons and social and solidarity spheres can explain that.

In sum, the present study's findings highlight the need for democratic governance economic models rather than data, knowledge and technological ones. The findings also demonstrate the relevance of the interconnections among governance, economic model and technological and data policy dimensions in promoting the collaborative economy. In addition, the results point to a disconnection between social responsibility and impact with the rest of the dimensions. Nevertheless, in future research, this part of the analysis could be repeated and validated to enrich the indicators and improve the potential correlations with the other dimensions. Future research could also aim to identify causal rather than correlational relationships among dimensions.

**Author Contributions:** Equal contribution.

**Funding:** This research is part of the work carried out by the Dimmons research group in the framework of the DECODE project (funded by the European Union's Horizon 2020 Programme, under grant agreement number 732546).

**Acknowledgments:** We would like to thank E. Senabre, E. Rosello, M. Rocas, G. Smorto, B. Carballa, P. Imperatore, M. Rebordosa, and N. Andrea for their contribution to the data collection, and C. Malpica for her help in data analysis.

**Conflicts of Interest:** The authors declare no conflict of interest. The funders had no role in the design of the study; in the collection, analysis, or interpretation of data; in the writing of the manuscript, or in the decision to publish the results.

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
