# Peer review of "A Framework to Assess the Sustainability of Platform Economy: The Case of Barcelona Ecosystem"

_sustainability, doi:10.3390/su11226450_

Round 1
Reviewer 1 Report
The authors are addressing an important issue. They produce a sustainability pro-democratic quality balance of the PE and test it in a sample of 100 PE’s in Barcelona.
The manuscript would be improved if the authors more effectively communicate:
Author Response
We would like to thank the reviewer for careful and thorough reading of this manuscript and for the thoughtful comments and constructive suggestions, which help to improve the quality of this manuscript. Our response follows:
We fully agree with the observation of the reviewer about the lack of social responsibility dimension, which it has been matched with the impact one. Indeed, this is the main challenge that we identified in our research (it is mentioned in the conclusions) and the aspect that we are currently working throughout a new research which take into consideration SDGs. In any case, we have tried to explain better. The whole manuscript references to have been reviewed. Finally, we have included the reference to the codebook to show the indicators used.Taking advantage of the message, we are attaching the manuscript reviewed.

Reviewer 2 Report
Provided that sustainability is a wide and complex topic, i think that this paper is interesting, as it analyses significant aspects and enlarges the knowledge on the topic. The main lack that I found concerns the environmental aspects. Environmental sustainability is here considered as a sub-section of Social responsibility, by introducing few and general indicators on circular economy (that represents a wide topic by itself) and environmental responsibility. I would have rather considered environmental sustainability as another dimension of the “star” reported in figure 1. This is just a suggestion in view of further developments of the study.
Similarly, I think that the word “ecosystem” is not used in an appropriate way in this paper. Ecosystems are systems in which there is a strong interaction between humans, species and the environment. Thus I would suggest to replace the word “ecosystem” with “system” everywhere.
I also suggest the following minor corrections:
Line 39: “2) There is confusion about the PE which presents them as collaborative,” this sentence is not clear, who is “them” referred to? Lines 247-256: This part is not clear. How was the selection of relevant cases done? (why 100 cases, then 1000 cases, and 50 platform managers?) Definition of all acronyms (LGPL, MIT, FLOSS etc.) should be provided Line 275. Please provide reference for Alexa and Kred Line 524 and others: what do the authors mean for “pro-democratic”? Lines 584-586: this conclusion should be better supported by the results Lines 591-593: same as previous pointAuthor Response
We would like to thank the reviewer for careful and thorough reading of this manuscript and for the thoughtful comments and constructive suggestions, which help to improve the quality of this manuscript. Our response follows:
The whole manuscript references have been reviewed. We have tried to clarify what is the sample election. Departing from a previous research (1000 cases) and performing web collection (in 100 cases) and phone collection (in 50 of the 100 cases selected). We have decided also to avoid the use of “pro-democratic” and change for “democratic". It is almost the same meaning and simplify the comprension. We have also reviewed the conclusions to support better the results. Regarding to the use of “ecosystem” we agree that even though initially refers to systems in which there is a strong interaction between humans, species and the environment, currently it is spread the use of it to refer to the interaction among different type of organizations or the interaction among information systems to name but two. Thus, the term "system" does not means what we are trying to support our approach about the elements that configure the platform economy in this case in Barcelona. Finally, we fully agree with the observation of the reviewer about the lack of environmental concerns. Indeed, this is the main challenge that we identified in our research (because the mention in the conclusions) and the aspect that we are currently working throughout a new research which take into consideration SDGs.Taking advantage of the message, we are attaching the manuscript reviewed.

Reviewer 3 Report
Dear authors.
The manuscript here presented is very good. Congratulations.
It presents a novel framework which allows to assess Platform Economy, an increasing business model. The manuscript therefore results in an interesting study applied to an area of knowledge of growing interest.
Due to the quality of the document, few comments are necessary. However, some points that need to be corrected, or that may help to improve if possible the final quality of the article.
Review some references: line 73, line 80. Correlation tables help in finding the related items, but maybe some graphic/figure expressing the most relevant relationships could help. I believe that it would be of special interest to give greater relevance to the usefulness of the study (maybe in Methods, maybe in Conclusions). Knowing the relationships that exist between the dimensions studied, what is it useful for?Author Response
We would like to thank the reviewer for careful and thorough reading of this manuscript and for the thoughtful comments and constructive suggestions, which help to improve the quality of this manuscript. Our response follows:
The whole manuscript references have been reviewed. In the conclusions we have tried to deep in the value of correlations and their value.Taking advantage of the message, we are attaching the manuscript reviewed.
